# Impact of Angiotensin-Converting Enzyme Inhibitors or Angiotensin Receptor Blockers on Acute Kidney Injury in Emergency Medical Admissions

**DOI:** 10.3390/jcm10030412

**Published:** 2021-01-22

**Authors:** Athanasios Feidakis, Maria-Rosa Panagiotou, Emmanouil Tsoukakis, Dimitra Bacharaki, Paraskevi Gounari, Petros Nikolopoulos, Katerina P. Marathias, Sophia Lionaki, Demetrios Vlahakos

**Affiliations:** 12nd Department of Internal Medicine, National and Kapodistrian University of Athens Medical School, Attikon University Hospital, 1 Rimini Street, Haidari, 11527 Athens, Greece; th.fidakis@gmail.com (A.F.); mropanagiotou@gmail.com (M.-R.P.); manos1994osfp@hotmail.com (E.T.); bacharaki@gmail.com (D.B.); pari_gounari@yahoo.com (P.G.); nikolopoulospetros@gmail.com (P.N.); sofia.lionaki@gmail.com (S.L.); 2Intensive Care Unit, Onassis Cardiac Surgery Center, 17674 Athens, Greece; katerina@marathias.gr

**Keywords:** acute kidney injury, angiotensin-converting enzyme inhibitors, angiotensin receptor blockers, emergency medical admission

## Abstract

Background: Acute kidney injury (AKI) has been observed in up to 20% of adult hospital admissions. Sepsis, diarrhea and heart failure, all causing reduced effective volume, are considered risk factors for AKI, especially among patients treated with medications that block the Renin-Angiotensin System (RAS), such as angiotensin-converting enzyme inhibitors (ACEi) and angiotensin receptor blockers (ARB). We aimed to determine the incidence of acute kidney injury (AKI) in emergency medical admissions in relation to the use and dosage of ACEi/ARB. Methods: A single-center observational study conducted in 577 consecutive medical admissions via the Emergency Room (ER) at a University General Hospital in Athens, Greece, between June and July 2018. Patients with incomplete medical records, discharged within 24 h, maintained on chronic renal replacement therapy, admitted to the Cardiology Department or the ICU were excluded. Thus, a total of 309 patients were finally included in this analysis. Results: We compared 86 (28%) patients who presented in the ER with AKI (AKIGroup) with 223 (72%) patients without AKI (non-AKI Group) at the time of admission. Patients in the AKI Group were more frequently male (58% vs. 46%, *p* = 0.06), with a higher frequency of diarrhea (16% vs. 6%, *p* = 0.006), edema (15% vs. 6%, *p* = 0.014) and lower systolic blood pressure (120 (107–135) vs. 126 (113–140), *p* = 0.007) at presentation, despite higher prevalence of hypertension (64% vs. 47%, *p* = 0.006). Overall, ACEi/ARB were more likely to have been prescribed in the AKI Group than in the non-AKI Group (49% vs. 28%, *p* = 0.001). Interestingly, AKI was more frequently observed in patients treated with the target or above target dosage of ACEi/ARB, but not in those receiving lower than the recommended dosage. Conclusion: The risk of AKI in emergency medical admissions is higher among users of ACEIs/ARB at target or above target dosages. Physicians should adjust RAS blockade according to estimated Glomerular Filtration Rate (eGFR) and advise their patients to withhold ACEi/ARB in cases of acute illness.

## 1. Introduction

Acute kidney injury (AKI) is a rapid deterioration in kidney function, associated with increased mortality, prolonged hospital stay and the risk of chronic kidney disease [1,2]. AKI has been observed in up to 20% of adult hospital admissions [1,3,4]. Sepsis and diarrhea, causing reduced effective volume, are considered risk factors for AKI, especially in patients taking medications that block the Renin-Angiotensin System (RAS), such as angiotensin-converting enzyme inhibitors (ACEi) and angiotensin receptor blockers (ARB) [5].

Activation of RAS has been shown to be associated with long-term detrimental cardiorenal consequences. Consequently, ACEi/ARB are considered the cornerstones of treatment in patients with hypertension, diabetes, chronic kidney disease (CKD) or heart failure (HF) and their use has become increasingly widespread, driven by evidence-based guidance [6]. The beneficial effects of this intervention in various clinical trials are usually restricted to participants receiving higher dosages of RAS blockers. As a rule, high target doses of ACEi/ARB are well tolerated in patients with normal cardiac function and preserved renal perfusion. However, in many clinical studies, even after exclusion of patients with severe renal dysfunction, a significant fraction of patients did require a reduction of the post-randomization dosage, mostly due to older age, hypotension, hyperkalemia and deterioration of renal function. In a real-world situation, the number of patients who will have to receive a lower than optimal RAS blockade would probably be even higher [7,8].

Although ACEi/ARB are not considered to be nephrotoxic per se, they have long been regarded as a cause of AKI in patients with dehydration, cardiovascular decompensation or infections [9]. Indeed, the glomerular capillary pressure, and hence glomerular filtration rate (GFR), depends on the selective constriction of the efferent arteriole by angiotensin II; therefore, a rigorous RAS blockade in patients with low effective volume may be detrimental to renal function [10]. As a result, despite limited evidence from randomized studies, many suggest educating patients about “sick-day rules” for stopping ACEi/ARB in cases of acute conditions predisposing to AKI, such as fever, diarrhea, vomiting and excessive sweating [11]. However, this does not seem to be the case due to lack of patient education for various reasons. First, physicians may fail to properly educate their patients due to time constraints, busy clinics, burnout and even emotional exhaustion. On the other hand, patients may fail to understand instructions, due to language or cultural barriers, low socioeconomic and educational background and deficits in their cognitive status. Our goal was to examine the impact of taking ACEi/ARB in various doses on the development of AKI among patients admitted to our hospital for acute medical illness via the Emergency Room (ER).

## 2. Material and Methods

### 2.1. Study Design

This was a single-center observational study in 577 consecutive self-referring patients at the Emergency Room (ER) who were subsequently admitted to the Division of Medical Services at Attikon University Hospital in Athens, Greece, between June and July 2018. After excluding patients with unknown baseline creatinine, end-stage renal disease maintained on chronic renal replacement therapy, incomplete history (e.g., patients with dementia without caregiver at presentation) or medical records, discharged during the first 24 h of hospitalization, and admitted to the Intensive Care Unit or to the Cardiology Department, a total of 309 medical admissions constituted the final study population. The study protocol was approved by the ethics committee in Attikon University Hospital, Athens, Greece and was conducted in accordance with the World Medical Association Declaration of Helsinki. Additionally, upon the authors’ request, the committee waived the requirement to obtain informed consent for this study, since it involved secondary analysis of existing data and the privacy of subjects was protected.

### 2.2. Study Population and Acute Kidney Injury (AKI) Assessment

A detailed review of the patients’ history and previous medical records was conducted for demographic and clinical data, such as age, gender, body mass index (BMI), systolic and diastolic blood pressure measurements, heart rate, oxygen saturation and the presence of edema or hypovolemia. The chief complaint and the primary diagnosis upon admission were recorded. We also recorded the presence of diarrhea, vomiting, fever and heart decompensation within the past few days, which are known to compromise a patient’s hemodynamic status and further increase the risk for AKI.

Past medical history was reviewed for arterial hypertension, diabetes mellitus, dyslipidemia, hyperuricemia, coronary artery disease, congestive heart failure and arrhythmias, chronic kidney disease (CKD), respiratory diseases, malignancies, autoimmune diseases, gastrointestinal tract diseases, genitourinary diseases, hematopoietic system disorders and neurological diseases. The medical regimen for ACEi/ARB, β-blockers, diuretics, calcium channel blockers, metformin, insulin, allopurinol, anti-inflammatory agents, antiplatelet therapy, and food supplements was recorded. Laboratory data on admission included complete blood count, C-reactive protein, electrolytes and other biochemical markers, such as BUN, uric acid, creatinine, and glucose. CKD was defined as a baseline steady-state eGFR calculated by the MDRD equation of less than 60/mL/1.73 m^2^ on at least two occasions separated by 90 days [12].

AKI was determined according to the Acute Kidney Injury Network (AKIN) criteria, and patients were considered to have AKI when the serum creatinine measured at the ER was 1.5 times or more above their baseline creatinine. Baseline serum creatinine was measured within 7–365 days prior to hospital admission and was obtained from the hospital system or by data presented by the patients themselves.

### 2.3. Treatment with RAS Blockers

To be considered under treatment with RAS blockers, patients should have been given ACEi/ARB for at least one month before admission. Data were collected from the history report obtained by the ER physician and confirmed by the patient’s medical records dating back at least one year. Table 1 summarizes the suggested target dose for each agent and the proposed adjustment according to baseline renal function by manufacturers’ recommendations and clinical guidelines [13]. Accordingly, our patients on ACEi/ARB were categorized into three subsets according to ACEi/ARB dosage: 31% received “below target” dosage, 55% received “target” dosage and 14% received “above target” dosage. Interestingly, 13% of patients with CKD were prescribed a dose of RAS inhibitors not adjusted for their baseline renal function.

### 2.4. Statistical Analysis

Statistical analyses were performed using IBM SPSS statistical software version 25.0 for Windows (SPSS, Chicago, IL, USA). Continuous variables were tested for normality with the Shapiro-Wilk test and expressed as mean ± standard deviation or medians with interquartile range (IQR). Categorical variables were presented as frequencies and percentages. For the between-group comparisons, a Chi-square test was performed for categorical data and Kruskal-Wallis or Student t-tests were performed for continuous data with nonparametric or parametric distribution, respectively. A careful selection process was applied in order to choose confounding variables used in the logistic regression model. Variables that were statistically significant (*p*-value < 0.1) in the univariate logistic regression analysis and variables with clinical relevance to the study outcomes were introduced into the model. Specifically, adjustments were performed for the following variables: gender; diarrhea; systolic arterial pressure (mmHg); diastolic arterial pressure (mmHg); presence of edema; hypertension; dyslipidemia; CKD; the intake of β-blockers, diuretics, calcium channel blockers (CCBs), allopurinol and RAS blockers; and laboratory data of statistical and clinical significance: K, Na, Ca, P, BUN, uric acid and creatinine on admission. Multivariable logistic regression analysis was then performed in order to determine independent risk factors for AKI. The results were expressed as hazard ratios (HR) and 95% confidence intervals (CI). A two-tailed *p*-value of less than 0.05 was considered statistically significant.

## 3. Results

### 3.1. Baseline Characteristics and Risk Factors for AKI

During the observation period, a total of 577 patients were admitted to the Department of Internal Medicine via the ER. After excluding 268 patients using the criteria mentioned above, a total of 309 patients formed the study population and were finally analyzed. The mean age was 75 (60–83) years, and half of them were male. Patients were divided into two groups on the basis of AKI diagnosis upon presentation at the ER: 86 patients (28%) had AKI (AKI Group) and formed the study group, and 223 patients (72%) had stable renal function at presentation (non-AKI Group) and served as controls.

Table 2 reviews demographic and clinical characteristics on the day of admission via ER. In comparison with the non-AKI Group, patients in the AKI Group were more frequently male (58% vs. 46%, *p* = 0.06), had lower systolic arterial blood pressure at presentation [120 (107–135) vs. 126 (113–140) mmHg, *p* = 0.007] despite a more common past medical history of hypertension (64% vs. 47%, *p* = 0.006), a higher prevalence of dyslipidemia (31% vs. 17%, *p* = 0.008) and presumably a lower effective volume, as revealed in the physical examination and indicated by a three-times higher frequency of diarrhea (16% vs. 6%, *p* = 0.006) and edema (15% vs. 6%, *p* = 0.014). Table 3 summarizes laboratory data on admission. As expected, patients in the AKI Group had higher blood urea nitrogen (BUN), creatinine, uric acid, phosphate levels and lower calcium concentration.

Of note, as shown by the multivariate analysis (Table 4), the risk of AKI was almost three times higher in patients who presented with diarrhea compared to their counterparts without diarrhea (OR 2.86, *p* = 0.034). Patients in both groups were comparable in mean age, weight and height (BMI), history of coronary artery disease, heart failure, CKD, chronic obstructive pulmonary disease (COPD) and cancer.

### 3.2. RAS Blockers and AKI

As shown in Table 2, overall, RAS blockers were the most prescribed medications (34%), followed by β-blockers (33%) and diuretics (31%). Compared to patients nottreated with ACEi/ARB, patients receiving RAS blockers had, as expected, a higher prevalence of arterial hypertension (88% vs. 33% *p* < 0.001), diabetes mellitus (39% vs. 22%, *p* < 0.01), dyslipidemia (34% vs. 15%, *p* < 0.001), hyperuricemia (15% vs. 3%, *p* < 0.001) and COPD (14% vs. 6%, *p* = 0.019).They were also more likely to receive β-blockers (44% vs. 28%, *p* = 0.04), diuretics (53% vs. 20%, *p* < 0.001), CCBs (33% vs. 11%, *p* < 0.001) and metformin (24% vs. 7%, *p* < 0.001).

As shown in Figure 1, ACEi/ARB were prescribed more often in the AKI Group (49%) than in the non-AKI Group (28%) (*p* = 0.0004), and there was a dose-response effect with regard to rigorousness of RAS blockade. Thus, patients in the AKI Group compared to the non-AKI Group were more often given ACEi/ARB at an “above target” dosage (9% vs. 3%, *p* = 0.024) or at a “target” dosage (30% vs. 14 %, *p* = 0.003). Patients at a “below target” dosage of ACEi/ARB did not differ between the two groups (10% vs. 10% *p* = 0.97). In the final multivariate model (Table 4), those relations retained statistical significance. Patients on RAS blockade at “target dosage” had an estimated 2.94-fold increase in the risk for AKI (*p* = 0.013), while patients at an “above target” dosage had an estimated 4.85-fold increase in the risk for AKI (*p* = 0.025), in comparison to patients without RAS blockade.

## 4. Discussion

The overall incidence of AKI in patients admitted to the Department of Medicine for acute illness via the ER in this study and the patients’ characteristics compare closely with reports by others [12,14]. Our results suggest that RAS blockade is associated with increased risk of AKI. Moreover, patients treated with ACEi/ARB at the target or above the target dosage had a 3–5-times higher risk for AKI on admission, but no extra risk was shown if patients received lower than the recommended dosage of ACEi/ARB, in agreement with previous observations. For example, in the VA Diabetes in Nephropathy Study (VA-NEPHRON-D), the risk of AKI significantly increased when a more intense RAS blockade with a combination of ACEi and ARB was administered [15]. Mansfield et al. also showed that treatment of hypertensive patients with ACEi/ARB was associated with a small increase in AKI risk [16].

In addition to RAS inhibition, hypovolemia, hypotension and decompensated heart failure could have contributed to the occurrence of AKI by causing a reduction in effective volume. In this regard, our patients with AKI had more often diarrhea, peripheral edema and lower blood pressure at presentation despite higher prevalence of hypertension. Others have also shown a substantially increased risk of hospital admissions with AKI following periods of gastroenteritis in the community. However, in contradiction to our results, they found that the risk of AKI was similar in those using ACEi/ARB and those on other antihypertensive medications [5]. Tomlinson et al. found that up to 15% of the increase in AKI admissions in England over a 4-year time period were potentially attributable to an increase in prescriptions of ACEi/ARB [17]. In a nested case-control study, a double therapy combination containing diuretics or RAS inhibitors with NSAIDs was not associated with an increased rate of AKI. In contrast, current use of a triple therapy combination was associated with an increased rate of AKI [18]. There are many reasons to explain the divergent results between these studies. First, there is unavoidable variability in clinical characteristics among patients retrospectively selected for a study. When the sample size is relatively small, the likelihood of type II statistical error is higher. Additionally, the discrepancy of the results may be attributed to different populations and ethnic groups, different dosages of ACEi/ARB or diuretics used and different statistical models.

Although ACEi/ARB are not considered to be nephrotoxic per se, they have long been regarded as a cause of AKI in patients with concurrent dehydration, cardiovascular decompensation or infections [9]. As a result, despite limited evidence from randomized studies, there is a growing consensus to withhold RAS blockade during acute illness. The glomerular capillary pressure, and hence glomerular filtration rate (GFR), depends on the constriction of the efferent arteriole by angiotensin II; therefore, a rigorous RAS blockade in patients with low effective volume due hypovolemia, hypotension or severe HF may be detrimental to renal function [10,19]. For that reason, the results of the SPRINT study deserve special consideration. This study included nondiabetic patients over 50 years of age with increased cardiovascular risk, who were randomized to intensive or standard antihypertensive therapy. Even among participants without CKD at baseline, development of CKD occurred 3–4 times more frequently, and AKI occurred 2 times more frequently in the intensive treatment group than in the standard treatment group. These adverse renal outcomes seem to be related to an intraglomerular hemodynamic effect due to a more liberal use of diuretics (67% in the intensive treatment group vs. 43% in the standard treatment group) and of ACEi or ARBs (76% in the intensive treatment group vs. 55% in the standard treatment group) [20].

As a result, treatment with ACEi and ARB is widely believed to be a risk factor for AKI, particularly during acute illness. This underlies the “sick-day rules” recommendation for patients to stop taking these drugs when they become acutely unwell with symptoms of gastroenteritis or fever [21]. This study was not designed to examine the effect of temporary cessation of these drugs on the development of AKI. However, the fact that our elderly hypertensive patients on a lower than recommended target dosage of RAS blockade did not have higher risk for the development of AKI upon their presentation in ER may suggest that an adjustment of the dose for elderly individuals with reduced GFR is warranted, and reinforces the importance of assessing GFR and not relying only on serum creatinine in planning to prescribe ACEi/ARB. Interestingly, 70% of our patients with baseline CKD (eGFR < 60 mL/min at their steady-state condition) were unaware of having CKD; 80% had moderate stage III CKD and 20% had severe stage IV–V CKD.

We acknowledge that there are several limitations in this study: first, severely ill patients who were admitted to the ICU or the Cardiology Department and relatively healthy individuals who were discharged home during the first 24 h were excluded from this analysis. Therefore, the sample size remained relatively small and did not allow us to perform subgroup analyses for the effects of ACEi/ARB on AKI. Second, the definition of AKI was based only on changes in plasma creatinine level because urine output data were not uniformly available. Nonetheless, the strengths of the present study include the strict criteria for the inclusion of cases in a single teaching university hospital, the use of well-validated automatic analyzers in a modern certified clinical laboratory and the recording of a complete and detailed list of the type and dosage of medications taken on admission.

In conclusion, patients admitted to the Department of Internal Medicine via the ER for acute illness have higher risk for AKI if they had been previously treated with ACEi/ARB at the target or above the target dosage. No increased risk was seen in patients treated with ACEi/ARB at a lower-than-target dosage. In addition to RAS blockade, hypovolemia, hypotension and decompensated heart failure may have contributed to AKI by causing a reduction in effective volume. Physicians should be trained to adjust RAS blockade according to eGFR and advise their patients to withhold ACEi/ARB in cases of acute illness.

## Figures and Tables

**Figure 1 jcm-10-00412-f001:**
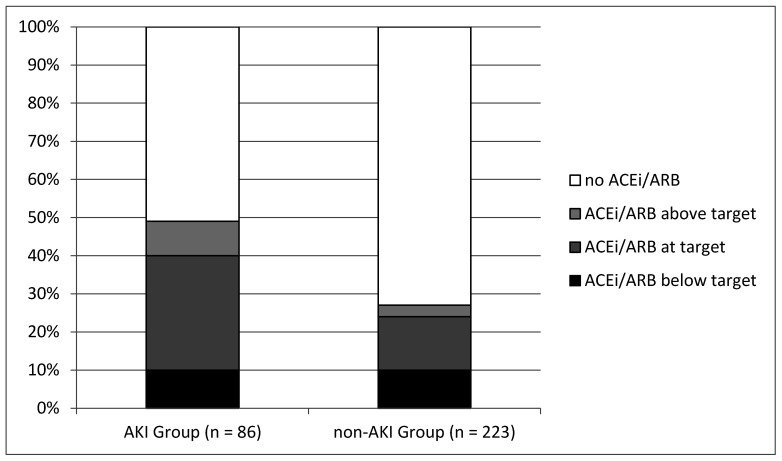
Percentage of patients on ACEi/ARB at above target, target and below target dosage in the AKI Group and in the non-AKI Group.

**Table 1 jcm-10-00412-t001:** Suggested dosage of ACEi and ARB and dose adjustment according to the GFR level.

Drug	Indication	Target Dose	Maximal Dose	Dose Adjustment	Excretion (Renal/Hepatic)
**ACEi**					
**Enalapril**	HF, HTN	10–20 mg/day	40mg/day	CrCl 30–80 mL/min: 5 mg/day CrCl 10–30 mL/min: 2.5 mg/day	100%/0%
**Perindopril**	HF	2.5 mg/day	5 mg/day	CrCl > 60mL/min: start 5 g/day CrCl 31–60 mL/min: start 2.5 mg/day CrCl 15–30mL/min: start 2.5 mg on alternate days CrCl < 15mL/min: start 2.5 mg/day on the day of dialysis	100%/0%
	HTN	10mg/day	10 mg/day		
**Ramipril**	HF, HTN	5 mg/day	10 mg/day	CrCl < 40 mL/min: start 1.25 mg QD, max 5 mg/day caution in elderly and hepatic impairment	100%/0%
**ARB**					
**Candesartan**	HF, HTN	32 mg/day	32 mg/day	If renal or hepatic impairment: start 4 mg/day	33%/67%
**Irbesartan**	HF, HTN	300 mg/day	300 mg/day	No adjustment recommended	30%/70%
**Losartan**	HF	150 mg/day	150 mg/day	CrCl < 20mL/min: 25 mg QD caution if hepatic impairment	10%/90%
	HTN	50 mg/day	100 mg/day		
**Olmesartan**	HF, HTN	20 mg/day	40 mg/day	No adjustment recommended	40%/60%
**Valsartan**	HF	160 mg/day	320 mg/day	If mild-moderate hepatic impairment: max dose 80 mg/day	30%/70%
	HTN	80–160 mg/day			

ACEi—Angiotensin Converting Enzyme inhibitors; ARB—angiotensin receptor blocker; HF—heart failure; HTN—hypertension; GFR—glomerular filtration rate.

**Table 2 jcm-10-00412-t002:** Comparison of demographic and clinical characteristics of the emergency medical admissions according to the development of AKI.

Characteristic *N* (%) or Median (IQR)	Overall (*n* = 309)	AKI Group (*n* = 86)	Non-AKI Group (*n* = 223)	*p* Value
Age (years)	75 (60–83)	73.5 (61.5–81)	76 (60–84)	0.810
Gender (Male)	153 (50)	50 (58)	103 (46)	0.060
BMI (kg/m^2^)	25.4 (21.7–28.7)	26.2 (22.1–29.4)	25.1(21.5–28.4)	0.320
Blood Pressure (mmHg)				
Systolic	125 (110–137.5)	120 (106.7–135)	126 (113–140)	0.007
Diastolic	70 (64–80.5)	70 (60–79.3)	70 (64–82)	0.076
Heart Rate (Beats/min)	80 (69–90)	81 (70–91)	80 (68–90)	0.650
Saturation O_2_ (%)	96 (94–98)	97 (94–98)	96 (93–98)	0.166
**Chief Complaint**				
CNS	88 (28)	23 (26.7)	65 (29.1)	0.675
Cardiovascular	12 (3.9)	2 (2.3)	10 (4.5)	0.379
Gastrointestinal	70 (22.7)	20 (23.3)	50 (22.4)	0.875
Respiratory	30 (9.7)	4 (4.7)	26 (11.7)	0.062
Musculoskeletal	14 (4.5)	5 (5.8)	9 (4)	0.501
Metabolic/Electrolyte	20 (6.5)	7 (8.1)	13 (5.8)	0.460
disturbances				
Hematopoietic	17 (5.5)	6 (7)	11 (4.9)	0.480
Genitourinary	28 (9.1)	13 (15.1)	15 (6.7)	0.021
Other	30 (9.7)	6 (7)	24 (10.8)	0.314
**Clinical Presentation**				
Diarrhea	28 (9.1)	14 (16)	14 (6)	0.006
Vomiting	45 (14.6)	15 (17.4)	30 (13.5)	0.370
Edema	27 (8.7)	13 (15)	14 (6)	0.014
Fever	62 (20.1)	21 (24.4)	41 (18.4)	0.396
Respiratory infection	27 (8.7)	5 (5.8)	22 (9.9)	0.260
Urinary tract infection	20 (6.5)	3 (3.5)	17 (7.6)	0.180
**Past Medical History**				
Hypertension	159 (51.5)	55 (64)	104 (46)	0.006
Diabetes Mellitus	86 (27.8)	26 (30.2)	60 (26.9)	0.560
Dyslipidemia	66 (21.4)	27 (31)	39 (17)	0.008
Hyperuricemia	22 (7.1)	9 (10.5)	13 (5.8)	0.160
CAD	36 (11.7)	12 (14)	24 (10.8)	0.430
Heart Failure	27 (8.7)	6 (6.9)	21 (9.4)	0.496
CKD	77 (24.9)	27 (31.4)	50 (22.4)	0.100
COPD	28 (9.1)	10 (11.6)	18 (8.1)	0.340
Cancer	55 (17.8)	18 (20.9)	37 (16.6)	0.370
Autoimmune Disorders	8 (2.6)	3 (3.5)	5 (2.2)	0.540
**Medications**				
ACEi/ARB	104 (34)	42 (49)	62 (28)	<0.001
-Above target dosage	15 (4.8)	8 (9)	7 (3)	0.024
-Target dosage	57 (18.4)	25 (30)	32 (14)	0.003
-Below target dosage	32 (10)	9 (10)	23 (10)	0.970
β-Blocker	103 (33)	35 (40.7)	68 (30.5)	0.089
Diuretic	97 (31)	30 (34.9)	67 (30)	0.410
CCB	56 (18.1)	19 (22.1)	37 (16.6)	0.260
Metformin	39 (12.6)	12 (14)	27 (12.1)	0.660
Insulin	26 (8.4)	8 (9.3)	18 (8.1)	0.720
Allopurinol	31 (10)	13 (15.1)	18 (8.1)	0.070

ACEi—ACE inhibitors; ARB—angiotensin receptor blocker; BMI—body mass index; CNS—central nervous system; CCB—calcium channel blocker; CAD—coronary artery disease; CKD—chronic kidney disease; COPD—chronic obstructive pulmonary disease.

**Table 3 jcm-10-00412-t003:** Laboratory data of the emergency medical admissions according to the development of AKI.

Value, Median (IQR)	Overall (*n* = 309)	AKI Group (*n* = 86)	Non-AKI Group (*n* = 223)	*p* Value
Hematocrit (%)	35.8 (30.95–40.35)	34.65 (29.55–38.53)	36.4 (31.4–40.7)	0.107
Hemoglobin (g/dL)	12 (10–13.5)	11.55 (9.68–13)	12.1 (10.3–13.6)	0.220
MCV (%)	86.7 (81.9–90.75)	86.5 (80.55–89.75)	86.8 (82–91)	0.950
Platelets (K/μL)	236 (173–310)	234 (168.8–309.3)	236 (174–311)	0.395
WBCs (K/μL)	9.08 (7.03–12.44)	9.01 (7.13–13.41)	9.1 (6.98–11.85)	0.180
CRP (mg/L)	21.1 (3.71–92.4)	34.7 (5.99–93.98)	16.3 (3.35–91.8)	0.130
Glucose (mg/dL)	115 (96–148)	119.5 (98–163.3)	112 (96–143)	0.040
Na (mg/dL)	139 (135–141)	137 (133.8–141)	139 (136–141)	0.070
K (mg/dL)	4.4 (4–5)	4.5 (4.05–5.4)	4.3 (4–4.9)	0.250
Ca (mg/dL)	9 (8.55–9.4)	8.9 (8.4–9.3)	9.1 (8.7–9.4)	0.020
P (mg/dL)	3.2 (2.7–3.7)	3.5 (2.9–4.25)	3.1 (2.6–3.5)	<0.001
BUN (mg/dL)	43.8 (30.15–65.55)	67.4 (44.5–110.1)	37.8 (26.1–52.5)	<0.001
Uric acid (mg/dL)	5.2 (3.95–7.25)	7.15 (5.38–9.28)	4.7 (3.7–6.3)	<0.001
Creatinine (mg/dL) on admission	0.9 (0.7–1.3)	1.4 (1.1–2.63)	0.8 (0.6–1.0)	<0.001
Baseline Creatinine (mg/dL)	0.7 (0.6–0.95)	0.8 (0.6–1.0)	0.7 (0.6–0.9)	0.386

WBCs—white blood cells; CRP—C-reactive protein; BUN—blood urea nitrogen; Na—sodium; K—potassium; Ca—calcium; P—phosphate; BUN—blood urea nitrogen.

**Table 4 jcm-10-00412-t004:** Univariate and multivariate logistic regression analysis.

	Univariate Analysis		Multivariate Analysis	
Characteristic	HR (95% CI)	*p* Value	HR (95% CI)	*p* Value
Age (years)	1.00 (0.99–1.02)	0.816		
Gender (Male)	1.61 (0.83–2.70)	0.061	1.85 (1.01–3.37)	0.520
BMI (kg/m^2^)	1.02 (0.98–1.07)	0.329		
Blood Pressure (mmHg)				
Systolic	0.98 (0.97–1.00)	0.007	0.98 (0.96–1.00)	0.010
Diastolic	0.99 (0.97–1.00)	0.077	1.00 (0.97–1.03)	0.931
Heart Rate (Beats/min)	1.00 (0.99–1.01)	0.657		
Saturation O_2_ (%)	1.00 (1.00–1.00)	0.767		
**Clinical Presentation**				
Diarrhea	2.90 (1.32–6.38)	0.008	2.86 (0.13–0.91)	0.034
Vomiting	1.36 (0.69–2.67)	0.374		
Edema	2.37 (1.14–4.92)	0.021	2.86 (0.02–7.12)	0.494
Fever	1.17 (0.88–1.55)	0.278		
Respiratory infection	0.56 (0.21–1.54)	0.264		
Urinary tract infection	0.44 (0.13–1.53)	0.197		
**Past Medical History**				
Hypertension	2.03 (1.22–3.39)	0.007	1.31 (0.34–1.70)	0.506
Diabetes Mellitus	1.12 (0.68–2.04)	0.559		
Dyslipidemia	2.16 (1.22–3.82)	0.008	2.03 (0.24–1.01)	0.054
Hyperuricemia	1.89 (0.78–4.60)	0.161		
Coronary Artery Disease	1.35 (0.64–2.83)	0.434		
Heart Failure	0.72 (0.28–1.85)	0.498		
CKD	1.58 (0.91–2.75)	0.104	1.31 (0.35–1.66)	0.490
COPD	1.50 (0.66–3.39)	0.332		
Cancer	1.33 (0.71–2.49)	0.373		
Autoimmune Disorders	1.58 (0.37–6.74)	0.540		
**Medications**				
ACEi/ARB	1.73 (1.30–2.29)	<0.001		
-Above target dosage	4.18 (1.44–12.16)	0.009	4.85 (1.22–19.33)	0.025
-Target dosage	2.86 (1.54–5.32)	0.001	2.94 (1.26–6.89)	0.013
-Below target dosage	1.43 (0.62–3.32)	0.402	1.42 (0.49–4.08)	0.519
β-Blocker	1.56 (0.93–2.62)	0.089	1.32 (0.38–1.52)	0.438
Diuretic	1.25 (0.74–2.12)	0.412	1.64 (0.79–3.38)	0.185
CCB	0.51 (0.30–0.89)	0.018	1.04 (0.42–2.17)	0.917
Metformin	1.18 (0.57–2.44)	0.662		
Insulin	1.17 (0.49–2.80)	0.727		
Allopurinol	2.03 (0.95–4.34)	0.069	1.07 (0.26–3.38)	0.921
**Laboratory Data**				
Hematocrit (%)	0.97 (0.94–1.01)	0.108		
Hemoglobin	0.94 (0.85–1.04)	0.220		
Na	0.97 (0.93–1.00)	0.072	0.99 (0.95–1.05)	0.978
K	1.38 (1.04–1.83)	0.028	1.02 (0.66–1.56)	0.946
Ca	0.67 (0.48–0.94)	0.019	0.62 (0.39–0.98)	0.058
P	1.91 (1.44–2.52)	<0.001	1.29 (0.85–1.95)	0.238
BUN	1.02 (1.01–1.03)	<0.001	1.01 (0.93–1.02)	0.392
Creatinine on admission	2.64 (1.88–3.72)	<0.001	1.82 (1.05–3.15)	0.033
Uric acid	1.44 (1.28–1.61)	<0.001	1.34 (1.14–1.56)	0.067

ACEi—ACE inhibitors; ARB—angiotensin receptor blocker; BMI—body mass index; CKD—chronic kidney disease; COPD—chronic obstructive pulmonary disease.

## Data Availability

Authors abide to the MDPI Research Data Policies.

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
