# Peer review of "Impact of Angiotensin-Converting Enzyme Inhibitors or Angiotensin Receptor Blockers on Acute Kidney Injury in Emergency Medical Admissions"

_jcm, 2021, doi:10.3390/jcm10030412_

Round 1

Reviewer 1 Report

Dear Editor,

The authors present an analysis of 577 admissions to the emergency room to evaluate if ACEi/ARB use is associated with development of acute kidney injury.   The study demonstrates that outpatient use of ACEi/ARB is associated with acute kidney injury upon presentation to the emergency room.   This aspect of the study is not necessarily novel.  The most important aspect of the study is the dose dependence.  I believe this manuscript is appropriate for publication with the following minor revisions.

  1. I would recommend including references for the ACEi/ARB dosing cutoffs used in the study.
  2. The authors needs to provide additional information in the statistical analysis. The authors stated that “A careful selection process was applied …”.  What was this “careful selection” process.  In addition, I would be helpful for them to include all of the variables in their logistic regression model in the text.
  3. It may be useful to look at number of anti-hypertensive agents used as well. I appreciate the ACEi/ARB use is the focus of the paper.
  4. Several patients are on combination therapy with ACEi/ARB + diuretic. How were the authors able to separate the effects of patients on combination therapy?

Reviewer 2 Report

In their retrospective study, Feidakis et al. investigate the association of RAS blockade with the incidence of AKI in patients presenting to the emergency room. The authors find that RAS blockade is more common in patients with AKI, hence posing a risk factor, particularly in the presence of other risk factors such as diarrhea.

The hypothesis of the study is clear, and the results seem to support this. However, the study has serious limitations.

1.It is a single-center study during the hot summer months in Athens which had probably an impact on the results. Therefore, it would be important to reproduce the findings in the same ER in a sample during the winter months.

  1. The definition of AKI was a 1.5 fold increase in serum creatinine. This is somewhat different than the definition of AKI according to the AKIN criteria in which an increase by 0.3 mg/dl defines AKI. The baseline creatinine is around 1 mg/dl so that this criterion should be applicable. What was the reason for this definition of a 1.5 fold increase?
  2. The authors should include the table with the multivariate logistic regression. How were the variables selected for the final model?
  3. In the introduction, the authors should point out the rationale for their study and the other studies in the field.
  4. The figure should contain the number of patients (n-number).
  5. Were the data normally distributed? In clinical studies, medians with interquartile ranges are more common.
  6. Table 3 should contain data on proteinuria and presence of other abnormalities (e.g. hematuria).

Round 2

Reviewer 2 Report

The authors have adequately revised their manuscript. I recommend its acceptance.